# Psychological Resilience of Volunteers in a South African Health Care Context: A Salutogenic Approach and Hermeneutic Phenomenological Inquiry

**DOI:** 10.3390/ijerph17082922

**Published:** 2020-04-24

**Authors:** Antoni Barnard, Aleksandra Furtak

**Affiliations:** 1Department of Industrial and Organisational Psychology, University of South Africa, Pretoria 0003, South Africa; 2Department of Human Resource Management, University of South Africa, Pretoria 0003, South Africa; hyraam@unisa.ac.za

**Keywords:** coping, general resistance resources, positive adaptation, psychological resilience, salutogenic, volunteering

## Abstract

Volunteering in non-Western countries, such as South Africa, is subject to poor infrastructure, lack of resources, poverty-stricken conditions and often conducted by volunteers from lower socio-economic spheres of society. Sustaining the well-being of volunteers in this context is essential in ensuring their continued capacity to volunteer. To do so, it is important to understand the psychological resilience of these volunteers and the resistance resources they employ to positively adapt to their challenging work-life circumstances. The aim of this qualitative hermeneutic phenomenological study was to explore volunteers’ psychological resilience from a salutogenic perspective. In-depth interviews were conducted with eight volunteers servicing government-run hospitals. Data were analysed through phenomenological hermeneutical analysis. Findings show a characteristic work-life orientation to be at the root of volunteers’ resilience. Their work-life orientation is based on a distinct inner drive, an other-directedness and a “calling” work orientation. It is proposed that this work-life orientation enables volunteers in this study context, to cope with and positively adapt to challenging work-life circumstances and continue volunteering. The elements of their work-life orientation are presented as intrapersonal strength resources fundamental to their psychological resilience. It is suggested that organisations invest in developmental interventions that endorse and promote these intrapersonal strengths.

## 1. Introduction

Volunteering is an essential and natural part of cultures across the world rendering significant multi-dimensional benefits to individuals, organisations and society [1,2,3,4]. Volunteers contribute to economic development and boost socio-economic phenomena such as social cohesion, citizenship, community development and social transformation [3,5,6,7]. Per definition, volunteering is an unpaid, planned, proactive helping activity where someone’s time, effort and energy is given freely for the benefit of other people, groups or organisations [8] to help solve social problems [9]. To continue volunteering over time, such commitment typically incurs personal costs, frequently under very difficult personal and economic circumstances [10,11]. Volunteers often operate in emotionally taxing environments, with limited organisational resources and inadequate training and they suffer stress and burnout [12,13]. Therefore, even though volunteering holds physical and psychological well-being benefits for the individual [2,14,15], these benefits may be compromised by the challenges volunteers face [16]. This may be especially relevant in an African context where volunteerism is typically constrained by poverty [6] inadequate training, poor support and lack of supervision, as well as logistical and financial limitations [17].

Volunteering infrastructure in non-Western countries such as South Africa is fast growing and promises to bridge the challenges that international volunteers face by volunteering in non-Western contexts [18]. For the 2017/2018 financial year, research with 74 leading South African companies showed that 80% of these companies run formal employee volunteer programmes and 46% employed designated full or part-time staff to manage volunteers [19]. Considering the high unemployment rate in the country, it is worthy to note that the reported 610.4 million volunteer hours in 2014, were equivalent to more than 293,000 full-time jobs, valued at R9.8 billion [20]. Volunteering in South Africa has played an important part in addressing key socio-economic and political challenges, yet lack of research and government support hamper its effectiveness [21]. The call for consistent research on volunteering in low income contexts [16] further underscore the value of such research in the South African context.

The study of volunteering in non-Western countries is important, however frequently relate to Western, international volunteers who come to Africa to volunteer [22,23]. Although this is similarly true of South African volunteering and native South African volunteers also stem from higher socioeconomic spheres of society, volunteering in the country is frequently conducted by disadvantaged people [6,24,25] who suffer the same physical and psychological health needs as the people who they care for [17]. It is this type of volunteer that stimulated the interest of the researchers because of the particular resource challenges they experience and have to cope with. There are many volunteers from less privileged backgrounds in the South African context. A study on volunteer characteristics in the country show that Black people volunteer more than double the hours that people from other population groups do and these volunteers report significantly lower levels of education than white and Coloured volunteers [24]. Of the Black volunteers in the study, 61.1% were unemployed and 37.6% of White volunteers were also unemployed [24]. Contrary to European, UK and American studies linking a higher level of education [26,27] and a higher social class and income [28] to volunteering, the 2014 South African volunteering activities survey (VAS) reported no relationship between hours spent volunteering and education and income [20].

Volunteer well-being in the work context is as important as that of paid employees [29,30] and understanding their coping resources and positive adaptation is beneficial to developing and sustaining volunteers [31]. In view of South Africa’s socio-political uniqueness, high unemployment and poverty rates, there is a need to conduct research on developing and managing the well-being of volunteers in this country’s context. South African volunteers working in high-risk medical care, further highlights a distinctly stressful and psychologically demanding work environment, with high performance demands, yet very limited support [32].

Psychological resilience is defined as a dynamic process of positive adaptation in the face of adversity [33,34]. Stressful work-life circumstances increase the risk for poor mental health, yet many people resile despite the difficulties they must endure [35]. Psychological resilience—the process whereby individuals maintain well-being despite adversity—is, among others, attributed to intrapersonal coping resources or positive psychological strengths that facilitate adaptive coping [35]. Salutogenesis originated as a stress and coping model [36] and is defined as a meta-theoretical paradigm focusing on the resources for health [37]; or a stress resistance resource approach emphasising one’s capacity to effectively apply available coping resources [38,39]. Central to salutogenic theory is the sense of coherence (SOC) construct, which is described as a wellness-protecting orientation to life that facilitates coping and positive adaptation in trying circumstances [40,41]. People with a strong SOC view life’s challenges as meaningful to engage with and believe that they have the ability to comprehend, manage and respond constructively to challenges [42]. These beliefs reflect the three SOC subcomponents of comprehensibility (cognitive component), manageability (behavioural component) and meaningfulness (motivational component). SOC plays a predominant role in promoting psychological resilience under stressful circumstances [43].

Another core construct in the salutogenic model namely generalized resistance resources (GRR) denotes person, group or environment characteristics that facilitate positive adaptation and coping despite stressful circumstances [36]. Generalized resistance resources play a dual role in positive adaptation. On the one hand they strengthen a person’s SOC and on the other they enable the use of specific resistance resources (SRR) in one’s immediate environment [44]. The aim of this study was to explore the psychological resilience of eight volunteers in a South African public health context from a salutogenic perspective. This study contributes to the body of knowledge by offering an in-depth understanding of the GRRs that strengthen volunteer resilience.

## 2. Materials and Methods

In this section, the research methodology and the research setting are provided, and the research methods are described in terms of sampling and participants, data collection, data analysis and ethical considerations.

### 2.1. Research Methodology

A qualitative study was conducted following a hermeneutic phenomenological approach and the epistemological notions of social constructionism. In this tradition, knowledge generation is based on the researcher’s interpretation of participants’ lived experience in a social context [45,46,47,48]. Findings of the study present the co-constructed meaning between the researcher and researched [49]. Findings do not claim a single or ultimate truth, but rather a perspectival, socially constructed meaning [46,50]. Such an approach is particularly appropriate to context specific research because meaning is derived from participants located in specific social and cultural contexts [51,52]. The hermeneutic agenda calls for critical interpretation by employing an established meta-theory in making sense of the research phenomenon [47,53]. The meta-theoretical orientation applied in this study pertain to the salutogenic perspective on well-being.

### 2.2. Research Setting

The study was conducted in a faith-based non-profit organisation (NPO) operating in 13 hospitals in the Gauteng and Western Cape provincial health sectors. The hospitals are government run; some situated in developed urban suburbs, and some in townships on the outskirts of a city. Government hospitals in South Africa are characterised by poor service delivery and hygiene, old and poorly maintained infrastructure and medical negligence [54]. The volunteers provide spiritual care and counselling, as well as emotional, social, trauma and physical support to patients and their families. Most of the volunteers come from poor communities and are faced with unemployment and poverty challenges.

Access to the research setting was gained through the management of the NPO, who provided written permission for the study to be conducted. A volunteer coordinator at the NPO was appointed as gatekeeper and assisted to identify and contact volunteers fitting the research inclusion criteria. Eight participants were contacted telephonically and informed about the nature of the study, the researcher was introduced as a psychologist and doctoral student, and they were requested to participate on provision of anonymity, confidentiality and their right to withdraw. All eight agreed to participate and interview logistics were arranged. Before proceeding with the interviews, each participant signed a consent form after perusing a participant information sheet, explaining the nature and purpose of the study as well as their rights as participants.

### 2.3. Researcher Roles and Preconceptions

The study became possible because of the second author’s involvement with her faculty’s community service project with the said NPO. At the time she was a lecturer in human resource management and a registered industrial and organisational psychologist with the Health Professions Council of South Africa (HPCSA). While the community service project focused on mentorship at the time, the second author’s interest in well-being, coping and retention of volunteers working at the NPO evolved into a research project for her PhD. This article forms part of her PhD and was co-conceptualised with her promoter, the first author. The first author holds a doctoral degree and she is a full professor in industrial and organisational psychology and a registered psychologist with the HPCSA in the categories of industrial and organisational as well as counselling. Both authors’ research interests focus on employee well-being in the workplace through in-depth qualitative inquiry. The second author conducted the interviews and both authors contributed to the methodology, data analysis and writing of this article.

### 2.4. Sampling and Participants

A convenient, purposive sampling strategy was employed to select information-rich participants [55,56]. Inclusion criteria were based on the definition of a formal volunteer which entails voluntary, non-paid service to others over an extended period through a formal organisation or agency [11]. Eight participants performing volunteer services through the NPO for 12 months and more, were invited and were interviewed in the period between April and July 2016. The eight volunteers, who each serviced one of four Gauteng-based hospitals constituted an adequate sample size for phenomenological research [55]. Table 1 below provides an overview of the participant profiles.

### 2.5. Data Collection

Eight initial in-depth interviews (60–90 min each) and three follow-up interviews (20–30 min each) were conducted. The interviews followed a flexible, thematic approach to elicit rich information by exploring the lifeworld of the participant [57]. After 11 interviews, data saturation was attained based on the depth (richness and thickness) thereof [58]. The in-depth interview allows flexibility to adjust thematically prepared questions during the interview, to facilitate a natural conversation flow and develop a narrative of lived experience, in which the research phenomenon is revealed [59,60]. To understand the antecedents that promote volunteers’ resilience and positive adaptation, the theme of the interview questions centred on the lifeworld experiences of volunteering as reflected in Table 2.

Two interviews were conducted on the premises of a district hospital. The other nine were done at the NPO head office, situated on the grounds of two public hospitals. Interviews were digitally recorded and transcribed by a professional transcriber. The software package Atlas.ti was used to store and manage the data.

### 2.6. Data Analysis

Data were analysed through phenomenological hermeneutical analysis entailing a naïve reading, constructing a structural thematic analysis and developing a comprehensive understanding [61,62]. Throughout the three stages, the metaphorical action of the hermeneutic circle is constantly applied, causing the researcher to move back and forth between the three stages, using each as a critical reflection and verification of the other [62].

The naïve reading entails repetitive reading of the transcriptions to get a sense of it as a gestalt [62] without any thematising [63]. Thereafter, to construct the thematic structural analysis, sections of meaningful text are identified and condensed in everyday language [62]. Condensed text is then reviewed, interpreted and clustered into sub-themes and main themes while continuously reflecting back on the naïve understanding and while constantly considering the research aim [61,62]. The comprehensive understanding is lastly constructed by reflecting on the holistic meaning in relation to the naïve reading, the thematic structural analysis, the research aim, the context of the study, the author’s preunderstanding and relevant meta-theoretical literature [62]. The findings reported below focus on the themes constructed in the structural analysis and the discussion that follows reflect the comprehensive understanding.

### 2.7. Ethical Considerations

Ethics approval was obtained from the relevant Institutional Ethics Committee (reference no. 2015_CEMS/IOP_050) and written permission to do the research was provided by the NPO in which the study was conducted. The study was conducted in line with the Ethics Policy of the University of South Africa (UNISA) and the Rules of Conduct for the Profession of Psychology of the HPCSA. Participants signed an informed consent prior to their participation. In reporting the results pseudonyms are used to ensure anonymity. Participant pseudonyms were used according to the abbreviation PR with the number of the participant following, for example, PR6 denotes participant six.

## 3. Findings

Based on the naïve reading, volunteers’ resilience seems strengthened by a distinct intrapersonal disposition or orientation to work and life. The structural thematic analysis conceptualises and synthesises this disposition or work-life orientation at the hand of three themes. The three themes describe how volunteers’ resilience is rooted in a characteristic inner drive, their other-directed life orientation and regarding their work as a ‘calling’. Next, each of the three main themes are conceptualised in sub-themes grounded in verbatim data.

### 3.1. Volunteer Resilience Rooted in a Unique Inner Drive

The volunteers’ unique inner drive is reflected in their self-determination and autonomous agency as well as in an innate desire to be productive and useful.

#### 3.1.1. Being Self-Determined and Demonstrating Autonomous Agency

The volunteers’ narratives reveal a characteristic self-determination and autonomous agency. They take responsibility for and are in control of their own lives and choices, as opposed to being directed by external forces, and this drives them to make their own decisions. Self-determination is demonstrated by PR4 who is active in shaping his own life and takes responsibility by acting persistently on his motives: “*…I became aware of the organisation, and then one day I came here, to XXX’s office… and then I came for a second time with the same person, and then I just decided okay, I want to continue with this, you know*”. Similarly, PR5 makes her internal locus of control apparent when she takes responsibility for her decision to volunteer: “… *you work under pressure, I am not working under pressure. When I am tired, or God wants to speak to me, I just stand and listen*”. Both PR4 and PR5 made a personal decision to become involved in volunteering. This personal and informed decision was based on a willing engagement that was free from external coercion.

Volunteer self-determination was not only evident in relation to the volunteering environment. PR1′s self-determination is revealed in the way she approached her life from an early age:
“You grow up knowing what you want and where you want to go. Because most of the children that I grew up with, their parents taking care of them and doing everything to them, today like they are still depending on their parents in such a way that everything the parents have to take decisions for them and even if a person is matured like me, they are waiting for their parents to take a decision for them… somebody has to come to a point where you have to take decisions on yourself”.

In addition to being self-determined, the volunteers are also autonomous agents, voicing a proclivity and capacity to make their own choices. This agency on the part of the volunteer is demonstrated by the free yet deliberate choices they made to engage in volunteer work. PR7 resolved: “*Volunteer is to work with your own ability, you do not, somebody does not push you. I want to volunteer, I want it*”. PR3 highlights that volunteering is “*a matter of choice*” and the deliberate choice to become involved in volunteering is confirmed by PR4 who indicates that volunteering “*is something you want to do*”.

This theme indicates the volunteer’s tendency to act independently, take deliberate action and apply freedom of choice. This innate predisposition of being self-determined and demonstrating autonomous agency acts as a general resistance resource, facilitating the volunteers’ resilience in vigorously continuing the work they do.

#### 3.1.2. A Desire to be Productive and Useful

Volunteers voice an innate desire to be busy and useful, despite their personal difficult circumstances and challenges. After retiring for health reasons, PR7 explains how she was not happy to sit at home and feel as if she was doing nothing: “*When I am at home I think about the patient because there are other patients there at the hospital, the patient who did not have the relatives, and they struggled a lot at the hospital and I decided to go there, not doing nothing*”. She (PR7) further emphasised that she could not sit at home knowing that she had the opportunity to contribute to the patients in the hospital: “*I do not want to sit at home doing nothing whereas there is somebody who want me to comfort her or him*”. Similarly, PR8 wanted to participate in new tasks as opposed to being inactive as a result of her health challenges: “*When I am busy staying at home I start thinking now, I am just sitting here, I do not do nothing… I start to think man, no man this sickness is going to kill me because I do nothing, I must start now, I’m going to rise up and …I must go and tell the people about something, encourage people at hospital*”. Despite her discomfort, PR8 is adamant that she needs to be productive explaining that she “*cannot sit here every day thinking of this pain, there is some other people there at hospital, they have got this pain also, I must go and say to him, no man God will help you, I must go and encourage the person*”. PR1 describes how she constantly strives to do something: “*I just made sure that all of my spare time I spend it in something, doing something. Either I am studying or I am helping somebody or I am doing something*”.

The volunteers’ need to be busy is complemented by the desire to be useful in their daily lives. This is evident from PR2, who lost her job after being declared incapacitated: “*… ek wil nou eerder in XXX, [met] sieklike mense gaan [tyd] spandeer as om by die huis te sit” (I would rather spend time with the ill patients than to sit at home)*. The extent of PR2′s desire to be useful is visible in the variety of activities she is actively involved in:
“Mondays I work with the SAP (South African Police Service) and when there is accidents or robberies or everything … Monday is this time. And when I am not busy, I pray for the people in the NPO... And then I—Tuesday is NPO. Wednesday is NPO. Thursday is ‘ouetehuis’ (old age home)—all the old people. And Friday is me and my husband come to NPO”.

In addition to the independent deliberate action and freedom of choice that acts as a resource for the volunteers’ coping and positive adaptation, their desire to be productive provides them with psychological resilience in that they are driven to take action, to be industrious and engaged in useful activity.

### 3.2. Volunteer Resilience Stemming from An Other-Directed Life Orientation

Volunteers’ other-directedness is characterised by being people-centred and having a religious orientation to life.

#### 3.2.1. Being People-Centred through Care, Compassion and Empathy

Volunteers unanimously report a strong people-centred orientation to life. This people-centredness is primarily rooted in an intrinsic desire to care for those in need. As described by PR1, this intrinsic need is inherent to her personality: “*I find that helping people is one of my, I don’t know if I can call it a weakness or a … because I can give what I have and remain with nothing by helping someone*”. This seems to suggest that the needs of others are more important than her own, illustrating how deeply ingrained and important this orientation is: “*I cannot live like this while others they are suffering outside. I rather use the small that I am having and do something*”. Similarly, PR3 indicates that he has a “*heart for people. I like you know helping people in a way that I can. Ja (yes) if it means buy you food I will buy you food*” and also explains that he is “*the type of person who sort of you know wants to do something for the people you know*” because “*there is something inside of me that needs to do good*”. Although a people-centred orientation to life is also inherent to PR3′s personality, he explains how it is further entrenched in him through his African values:
“Among the Africans, when somebody has lost a spouse or a child, then we go there and then, you know by going there it is the same as saying “Listen, I am here if there be any need, I am willing to get involved”, and they ask you to go and fetch water, they ask you to go and fetch wood and so forth. And during the circumstances of their mourning, then you provide some kind of help”.

Apart from an intrinsic need to care for others, having compassion is also central to the volunteer’s people-centred orientation to life. PR4 says that “*I am a person that have sympathy and empathy with other people, you know, in their time of suffering*”. PR5′s compassion, which is founded in experiencing her own suffering, drives her to ease the anguish of others spiritually: “*… in my heart there was a, I don’t know how to speak it, uh, a heart for that sick people because I come from there. I was so sick, I was feeling the love, to love them and show them that Jesus is the only way, there is no other way than Jesus*”. Similarly, PR8′s compassion motivates her: “*…but my spirit inside, I have got some, I have compassion with people, I want to encourage people with the words of God, you know*”. Being compassionate enables the volunteers to provide a support system for the hospital patients. This entails for example P6 acting as a family member who can listen to their fears and just be a presence next to their hospital bed. Likewise, PR4′s compassion drives him to encourage and listen to the patients and give them hope: “…*and sit down with them and just listen to them, listen to their fears or frustrations and things, and then you come there and you listen and you see in their eyes and you hear in their voice…and just give them hope*”.

Empathy also characterises the volunteer’s people-centredness and it stems from their own experiences. PR2 who underwent a back operation, explains how she is able to understand the pain, suffering and difficulty the patients are feeling and experiencing because of her own medical history: “*… when I get out of the 8 weeks, I will tell God ‘I know now how the people in the NPO, in the hospital feel’. Because I cannot tell you I know how you are feeling if you are not going through this*”. PR2 further mentions how she prays “*for everybody in this hospital, hospitals in the whole world because I know when I lay in the bed how they are feeling*”. Similarly, PR5 is also aware of what it entails to be a patient in the hospital: “*I come from there. I was so sick*”. Additionally, PR7 not only understands what it feels like to be a patient, but also how volunteer services were valuable to her: “*I was at hospital five years back, and when I was in hospital there comes a pastor and talk with me about God, and said God is love, and I take that message and restore my, my soul*”.

Being people-centred presents a unique other-directed orientation to life in general and presents as an intrapersonal strength or resource fundamental to the participants’ resilience in this volunteering context. The establishment of a disposition for well-being in the volunteer, such a people-centredness, is based on being driven by one’s care and compassion for others and the ability to feel empathy. The volunteer’s other directedness is also exemplified by its focus on a higher spiritual source, as discussed next.

#### 3.2.2. Religiously Rooted and Focused on God

A profound religious attitude, namely, a belief in and commitment to God, seems to be at the core of the volunteer’s positive adaptation. The volunteers’ focus on God is apparent as they have made it explicit that their effort is directed towards God himself and that they conduct volunteering as a service to Him: “*I work for God. I did not work for me, I work for God, who created me*” (PR7) and according to PR5, “*to volunteer, I don’t think it is a volunteer, it is just the work from God*”. PR5 also indicates that “*I’m going to the hospital for the purpose of God, for the sick people*”. PR2 further emphasises that “*I work for God, not for XXX, for God*”. Similarly, PR4 sees volunteering as a “*service in the kingdom of God*”.

The volunteers in this study were all specifically vocal about being directed by Christian principles and teachings. These engendered their serving behaviour, as explained by PR3: “*You know actually it’s one of the Christian principles where Jesus says, “If you want to be number one, start by serving… if you want to be number one, be a slave to everyone… I have been created to serve and to do good work*”. Similarly for PR6, to conduct service-oriented activities means to feel God’s power flowing through him: “*I know that if I want to be anointed I need to give something, so I need to give my service, that is why I am giving my service voluntarily*”. Christian practices and teachings furthermore direct the volunteers in their work role, as PR1 explains: “*There is something that I am relying on—it is the Bible. Most of the things that I do, I do according to the Bible*”. Acts of worship such as prayer and ministry provide patients with hope and encouragement as described by PR8: “*I am going to encourage them with the words of God and pray*”. PR2 recounts*: “…and I sit on the chair next to her and I pray for her and I give her one scripture, and I read in the Bible for her and… I ask her if I can lay my hands on her*”. While volunteering, the volunteers express the virtue of love, which for them is central to Christianity: “*I am trying to practise the Bible, that is what I have to give out to the patients…That love is to show them that there is a purpose for everything, there is an end out of everything*” (PR1). PR6 further explains that he tells the patients he loves them because “*when you start ministering you can see they need the love of God, because I believe Christianity is more about love than anything else*”.

Although the volunteer’s actions are predominantly focused on God and carried out in His service, signalling their commitment to Him, the volunteers also depend on God during times of difficulty, such as relying on Him for guidance to solve challenging problems faced in the workplace. This is noticeable in PR1′s explanation of dealing with a challenging patient while volunteering: “*You must ask God for a descending spirit that will help you to choose and to separate things and to do the right decision. So, sometimes when things like this are happening, I just ask God to help me how to come out of this or how to solve this*”. In this way, God acts as a support resource, fostering the belief that they will be able to deal with the demands posed by life. PR5 has faith that God will assist her with the challenges she is experiencing: “*I see my children, don’t have anything to eat or clothes, they said, mama (mom), we want money to go to school. I don’t have money, I am not working but I trust God. Because we are living by God’s grace, we are living by God’s grace*”. He (God) is further the source of the volunteer’s strength and gratitude. PR2 explains that “*when I feel down, and I can say “I am not so bad”, because the people in NPO in hospital is lying down, they don’t got legs, they have stomach cancer and everything you know. And then I say, ‘thank you God that I can make a difference, that you pick me up every morning’*”.

In addition to being people-centred in a caring, compassionate and empathetic manner, other-directedness is evident in being rooted in and deriving strength from a strong religious attitude, in this study context, specifically, a Christian religion.

#### 3.2.3. Work that is a “Calling”

The volunteer’s resilience is strengthened by a specific orientation to work, namely the need to conduct work that is regarded as a calling. This entails work that one is passionate about and intrinsically motived to do. Over and above their religious calling, this calling orientation to work specifically stood out in the narrative of PR1: “*being a volunteer I find that it is a calling. It is a calling and it is a passion*” which “*you will do it with passion and love*” and “*I found it being a calling like when you are called for something, when you are called for being a Pastor, when you are called to be a doctor, you are with that thing inside of you*”. PR3 shares this view: “*I think it’s a calling*”. A calling work orientation is described as being passionate about your work or having an intense love for it: “*a calling it is like something that you have a passion. Something that you have love when you do it. You have that, you do it whole heartedly with love. That is why you will be able to come here without somebody giving you an allowance, compensation to come here. You come here voluntarily. You come here using your own time for someone’s life*” (PR1). PR5 exemplifies her work passion by emphasising how much she “*love[s] this job. I love this job*”. This type of calling work orientation stems from an inner desire and motivation on the part of the volunteer to conduct such work and is reflected in the words of PR3: “*[Y]ou see to be a volunteer, it springs from the heart*”. In PR6′s explanation of how he started volunteering, he notes that volunteer work is intrinsically motivated: “*I wanted to volunteer, do the volunteer work but I never really have the volition, you know, if you know what I mean, I did not have the oomph to go and do it, because the passion was not ignited*” and how this changed for him to become a passion: “*but, from the time to 2012, when I came back from overseas, it was on my heart*”.

Two of the volunteers related to being called by God to volunteer, linking their work orientation to a religious calling. PR6 said: “*So ja (yes), it was not until that one day I had a dream, it was a vision actually, in the morning, and God said to me go to XXX…he said go and pray for the sick, and I went*”. PR2 had a similar experience during her recovery from a back operation: “And I got a voice from God, go to NPO and do something for the sick people because they are laying in the bed and they can do nothing for themselves”.

The volunteers’ resilience is supported by the unique way in which they approach their work environment, that is, from a calling work orientation. A calling orientation presents a unique orientation to work that reflects an approach based on being intrinsically motivated to do work one has a passion for.

## 4. Discussion

This article set out to explore volunteers’ psychological resilience from a salutogenic perspective. The findings highlight a characteristic predisposition or work-life orientation that supports and sustains positive adaptation as reflected in the will to continue volunteering.

In this study, volunteers from a low income context with limited material resources, portray a disposition or work-life orientation that is characterised by (i) a peculiar inner drive, (ii) an other-directedness and (iii) a “calling” work orientation. The three elements of their work-life orientation echo the sub-component dynamics of SOC and demonstrate how these dynamics are reflective of intrapersonal GRRs that promote and sustain their psychological resilience.

The first element of the volunteers’ predisposition is their inner drive. The volunteers’ unique inner drive is operationalised in their self-determined nature and autonomous agency as well as in a desire to be productive and useful. These intrapersonal characteristics are similar to volunteering studies that have linked autonomy orientation to engagement in pro-social activities, job satisfaction and intentions to sustain volunteer work [64,65]. Autonomy is also conceptually related to self-determination, independence, self-regulated behaviour and acting volitionally, according to one’s own will [66,67,68]. Self-determination and autonomy are furthermore important for optimal functioning and well-being [69,70] and, therefore, proposed here as important GRRs in the psychological resilience of volunteers in this study.

Volunteers’ inner drive reflects a predisposition to cognitively appraise difficult circumstances in a way that demonstrates a pervasive sense that life is comprehensible, manageable and meaningful. Their self-determination and autonomous agency support the SOC sub-component of *comprehensibility* in that volunteers take responsibility for their responses towards stressful stimuli in their external environment. Rather than blaming or questioning external forces, their appraisal of circumstances results in taking responsibility and acting decisively and of their own volition. Furthermore, the volunteers’ innate desire to be productive and useful is related to the SOC sub-component of *manageability*. Responding to their circumstances by taking action shows that the volunteers believe in their capacity to meet life’s demands. In being actively engaged and industrious through the volunteering work they do, volunteers confirm and build their self-efficacy and sense of feeling useful. This need to feel useful is fundamental to the volunteers’ orientation to be productive and to contribute, and therefore also relates to the motivational aspect of SOC namely, *meaningfulness*.

The second element in the volunteers’ predisposition, is their other-directedness. Having a life orientation directed towards servicing others or being in the service of religious beliefs and God, secondly also resemble the SOC components of *meaningfulness, manageability* and *comprehensibility*. In the data, their self-reflections and descriptions portray volunteers to be characteristically caring, compassionate and empathetic. This is congruent to studies showing that volunteers typically have pro-social personality characteristic such as other-oriented empathy and helpfulness which motivate them to volunteer [8]. The motivational effect of these characteristics is revealed in the dynamic that when enacting them, life is regarded as *meaningful*. Activities such as volunteering are therefore experienced as meaningful to engage in because the volunteer is then congruent to the authentic self. Their other-directedness also resemble the behavioural SOC sub-component of *manageability* as these characteristics provide intrapersonal resources that enable the volunteers to positively and actively respond to and act in their environment. Other-directedness thus seems to be a strength resource underlying psychological resilience in this study context, especially since expressing concern for the welfare of others, is directly related to psychological well-being [71] and empathy is considered a character strength [72]. Acting congruently to these pro-social traits, demonstrates volunteers’ cognitive appraisal of their circumstances as *comprehensible*, because they do not shy away from difficulties. They rather engage in life, despite its difficulties, by creating opportunities in which they can authentically enact their other-directed character traits and needs. Their other-directedness is also pertinently rooted in being religious and focused on God. Believing in God and enacting the religious call to serve others provide the volunteers with the potential for innate well-being, as it fosters *meaningfulness* strength resources such as hope and gratitude. Other-directed characteristics such as caring, compassion, empathy and a religious belief system thus act as intrapersonal strengths or GRRs that enable the volunteer to remain resilient in the face of adverse circumstances.

Volunteers’ psychological resilience is lastly rooted in the third element, namely a specific “calling” work orientation, which is conceptualised as having passion and an intrinsic motivation for the work. Volunteers with a religious identity (such as in this study context) have been shown to be motivated to volunteer as a way of following a calling [73]. Theory suggests that experiencing a calling to work results in positive effects such as work and life satisfaction, finding work meaningful, being more motivated and experiencing engagement with work [74]. Engaging in work that is aligned with a calling generally enhances well-being [75,76,77] and in this study context affirms the volunteers’ SOC through *meaningfulness*. Loving the volunteer work they do and being passionate about it, shows that the volunteers find meaning in answering the call to volunteer. The call to volunteer is, however, not only experienced in terms of their religion. Volunteering is experienced in general as a deep internal motivation to find purpose and meaning in life.

Research generally confirms that volunteering contributes to well-being [14,17,27,78,79,80]. This study contributes to the body of knowledge by explaining the dynamic that builds the psychological resilience of volunteers. The unique disposition of volunteers described in this study context, predisposes the volunteer to appraise and respond to stressful life circumstances in a way that builds their psychological resilience and leads to active coping and positive adaptation. Having GRRs, being aware of them and having the ability to use them buffer the risk of poor mental health and distress [81]. In this sense, the act of volunteering may be regarded as an SRR which the volunteer accesses in order to congruently enact their intrapersonal character strengths. Research generally cites the characteristics and values of volunteers as aspects that motivate their decision to volunteer. From this study, understanding how volunteers sustain their well-being despite the difficult circumstances they work and live in, augments the unique person characteristics that underscore the motivation to volunteer—not as motivational factors per se, but as their innate well-being potential. Developing an understanding of the unique intrapersonal GRRs that enable volunteers to resile, may enable organisations and government to better manage and retain this valuable resource.

The small sample and qualitative nature of the study present with in-depth, rich and contextual understanding, yet the study is limited with regard to generalisation. Moreover, the study had an idiographic purpose and did not investigate the potential GRRs present in the organisational or societal context of the study. Future research should explore the resources required to facilitate volunteer well-being and specifically investigate possible developmental interventions to promote volunteer resilience.

## 5. Conclusions

Due to their non-profit service agreement, volunteers do not receive the same benefits as full-time employees. Although organisations spend some resources on their recruitment, training and management, this return on investment is too often overlooked [4]. Investing in the development and well-being of volunteers seems to be an area of need, especially in African contexts that are subject to limited resources and lower socio-economic conditions of volunteers. The findings of this study have several implications for investing in volunteer well-being, based on the premise that volunteers will extend their services in the longer term if they constructively cope with and positively adapt to their limited work circumstances. Whether or not one volunteers or gets paid for your services, endorsing your character strengths is directly related to higher job and life satisfaction and indirectly to greater well-being [82]. It is, therefore, proposed that organisations employing the services of volunteers create developmental opportunities to identify and endorse volunteer character strengths such as their pro-social nature, inner drive and need to work with passion and purpose.

## Figures and Tables

**Table 1 ijerph-17-02922-t001:** Participant profiles.

Participant Acronyms.	Gender	Population Group	Age	Employment	Living Conditions	Hospital Situated in
PR1	Female	Black	34	Part time employment	Rural, low socio- economic upbringing	Developed urban area
PR2	Female	White	52	Unemployed	Lives in urban area. Receives financial support	Developed urban area
PR3	Male	Black	61	Self-employed	Rural, low socio- economic upbringing	Developed urban area
PR4	Male	White	58	Unemployed	Looks after sick mother. Lives in urban area	Developed urban area
PR5	Female	Black	42	Unemployed	Low socio-economic living circumstances	Township
PR6	Male	Black	60	Unemployed	Low socio-economic living circumstances	Township
PR7	Female	Black	58	Unemployed	Low socio-economic living circumstances	Township
PR8	Female	Black	53	Unemployed	Low socio-economic living circumstances	Township

**Table 2 ijerph-17-02922-t002:** Thematic interview questions.

Interview Questions
I would really like to know more about you. Can you tell me about your life story?
How did it come about that you started volunteering?
Can you tell me about your experiences of being a volunteer? What has happened since you started volunteering? What is it like to volunteer?

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
