# Peer review of "Psychological Resilience of Volunteers in a South African Health Care Context: A Salutogenic Approach and Hermeneutic Phenomenological Inquiry"

_ijerph, 2020, doi:10.3390/ijerph17082922_

Round 1

Reviewer 1 Report

The paper presents findings on the psychological resilience of the volunteers and how they adapt positively to the circumstances of working life through their intrapersonal strength. This relevant topic is focused on studying volunteering in non-Western volunteers.

Introduction

1.       Authors explain differences from the Western, international volunteer in terms of education, socio-economic class, social orientation and identity (page 2, line 45). Could you say more about the African volunteer picture with non-Western volunteers? Is it similar across the continent? As currently written, readers who might not know the context will be at a disadvantage. This will help to provide a stronger rationale for the study.

2.       The manuscript refers to the importance of volunteering in South-Africa. Although the topic is well defined, I suggest the authors reorganize the introduction section: firstly, defining what volunteering is, what volunteering on the African continent entails, who manages and performs the volunteering tasks in the context of the study. Then, define the characteristics of volunteers and identify external and internal strengths and weaknesses. In this way, the lack of studies on the subject can be reported more clearly. All this information has already been included by the authors, but I think that it should be presented in a clearer way.

3.       The main aim of the work or the specific hypotheses is not defined in the introduction section. Authors have included a sentence "The focus of this paper was, therefore, to explore the psychological resilience of volunteers in a South African public health context from a salutogenic perspective" (page 2, line 74) but are nor in the correct place.

Methods

Design

4.       Which author/s conducted the interview?

5.       What were the researcher’s credentials? E.g. PhD, MD

6.       What was their occupation at the time of the study?

7.       Was the researcher male or female?

8.       What experience or training did the researcher have?

9.       Was a relationship established prior to study commencement?

10.   What did the participants know about the researcher? e.g. personal goals, reasons for doing the research

11.   What characteristics were reported about the interviewer/facilitator? e.g. Bias, assumptions, reasons and interests in the research topic

Participants

12.   How were participants approached? How were potential participants identified?

13.   How many participants were in the study?

14.   How many people refused to participate or dropped out? Reasons?

15.   Can you add a statement that participants provided informed consent?

16.   Table 1 (page 3, line 131) could you clarify the meaning of the first column? P= patient, professional, person?

Data collection

17.   When was the data collected?

18.   Please, include sentence page 3, line 121, at the description of the sample section (participants)

Data analysis

19.   How was established data saturation?

20.   Were transcripts returned to participants for comment and/or correction?

21.   Please, include "ethical considerations" at the end of the methods section and before the findings section

Findings

22.   Was used any software to manage the data?

23.   Page 5, line 176: Could you clarify the meaning of PR4? Does PR4 = P4 in table 1?

24.   There is an inconsistency between the questions included in Table 2 (were not formulated to explore the psychological recovery capacity of the volunteers from a salutogenic perspective) and the findings presented in this article. There are no questions related to adaptation, own resources or coping despite stressful circumstances. As the authors explain at the discussion section “The findings extend the notion of researchers that individuals with particular characteristics select themselves into volunteer work, by demonstrating how these characteristics enable them to cope with and positively adapt to the extremely difficult circumstances they operate in.” (page 8, line 357) Did the authors decide to change the main objective once the study started, or was it perhaps an error in the design of the questions? Please, clarify and solve it.

Discussion

25.   Please, could you include in the discussion section, a brief summary of the main themes as ordered in the results and expose your ideas in a simple way. I think you have a good discussion but is difficult to follow due to the several concepts that appear. Try to simplify ideas.

26.   Page 9, line 380 could you firmly say that volunteers have an innate desire to be productive and useful, don't you think that could be mediated by internal motivation as religiousness, or external as previous experiences or educational process?

Reviewer 2 Report

It is an article that deals with an interesting topic. Below are a series of aspects to try to improve the article. Thanks for your attention:

The introduction does not express the number of participants or the percentage of women or men. It should appear.

The keywords should be in alphabetical order, to follow a specific order. It would also be interesting if one of the keywords referred to the type of study design.
It would be interesting not only to include a general objective but several clearly differentiated specific objectives.
Hypotheses associated with specific objectives could also be included.
Regarding the methodology, the number of subjects prevents robust statistical analysis. It is true that this is a qualitative study, but such a study would be richer and would contribute more if it had quantitative information. The design used, although interesting, makes it difficult for the study to be replicable.
Using the word "race" can be offensive in certain contexts. Another concept with better connotation and more politically correct would have to be studied. Example, in Table 1.
The rest of the sections develop the study with a good structure.
In summary, this is a topic of interest, but the design makes it difficult for the study to be replicable and for comparison with other studies. Furthermore, the number of participants is low. It is necessary to increase the number of participants and carry out a structuring in the design to include quantitative elements. Thank you very much for your attention and interest.

Round 2

Reviewer 1 Report

The manuscript has been significantly improved and now warrants publication in IJERPH. 

Congratulations!

Reviewer 2 Report

Thank you very much for the effort. I think a great job has been done. Congratulations. The manuscript deals with an interesting subject. I believe that the authors of the article have been able to defend and explain the number of participants, providing references and bibliography that demonstrate that it is possible to extract useful information and advance research. Thank you very much again for the effort.